# Plausibility based comprehension in a neural network model of sentence processing

## Abstract

Psycholinguistic evidence has shown that human language comprehension does not always proceed in accordance with syntactic rules. Instead, these rules can be overridden by semantic plausibility, challenging classic linguistic theories and models. Here we show that the phenomenon of plausibility based comprehension naturally emerges in the comprehension performance of the Sentence Gestalt model, a neural network model trained on mapping sentences to event description based on a large scale corpus without any explicit syntactic training.

## 1 Introduction

The meaning of a sentence is often assumed to be a function of the meaning of its constituent words and the thematic-roles assigned by morpho-syntactic cues and it is often assumed that sentence processing requires distinct processes such as lexical activation and syntactic parsing. Most theories assume that these processes unfold sequentially, that they are accurate, and that their respective output is detailed and complete. Over the past decades, this view has been challenged in psycholiguistics both by behavioral and electrophysiological evidences.

Ferreira (2003) asked human participants to indicate the agent or the patient of the event described by normal active sentences (e.g., "The dog bit the man"), role reversed active sentences (e.g., "The man bit the dog") and their passive versions. Role reversed sentences, often called reversal anomalies (RA) are sentences that are syntactically correct but semantically anomalous because their agent and patient fillers are swapped. In these sentences, the thematic-role assignment (e.g., "man" as *agent* and "dog" as *patient*) violates the expectations imposed by the event semantics which suggest that humans are more likely patients and dogs agents of a "biting" event. Ferreira's results showed that participants frequently misinterpret passive RA sentences

(e.g., "The dog was bitten by the man"). In consequence Ferreira (2003) proposed the "good enough" approach to language comprehension, which assumes that people might sometimes use processing heuristics based on their expectations about events to figure out who is doing what to whom rather than relying on syntactic rules. Relatedly, studies conducted by Kuperberg et al. (2003) and Kim and Osterhout (2005) show evidence that RA sentences, despite their semantic abnormality, elicit only a small increase in N400 amplitude compared to normal control sentences, which is surprising because amplitudes of the N400 brain potential are typically increased in semantically anomalous sentences (see Kutas and Federmeier 2011 for review). These observations were explained as the results of *semantic illusion* according to which the syntax-cued thematic-role assignment is - at least temporarily - overrun by expectations regarding the event semantics (Nieuwland and van Berkum, 2005), hence the small N400 amplitude. Both behavioral and electrophysiological studies therefore point to a (partial) overrule of syntactic information in favour of event-semantic priors when the thematic-role assignment appears to violate event probabilities.

In this paper we investigated whether the Sentence Gestalt (SG) model, a connectionist model of language comprehension that we trained on a large scale corpus, can account for the pattern of behavior elicited by RA sentences (active and passive), based on stimuli such as those used by Ferreira (2003) and Kuperberg et al. (2003). The SG model is a model of language processing that maps sentences to a representation of the described event approximated by a list of role-filler pairs representing the action, the various participants (e.g., agent and patient) as well as information concerning, for instance, the time, location, and the manner of the event described by the sentence itself (McClelland et al., 1989). Central to our simulation, is the fact

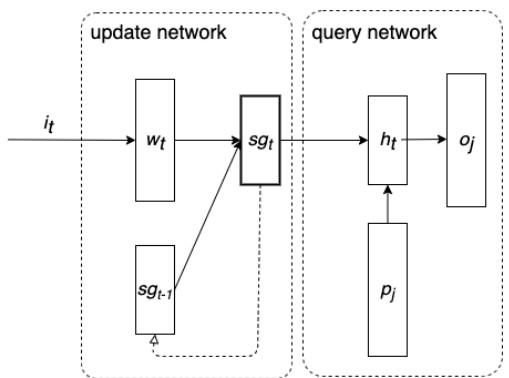

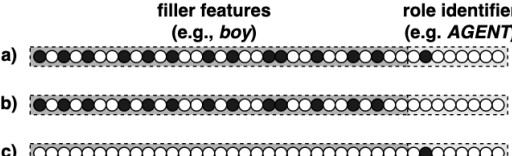

Figure 2: The role-filler vector $\vec{o}_i$ (a), and its corresponding two types of probes $\vec{p}_i$ (b) and (c). The left hand-side of the vectors correspond to the embedding representation of the filler concept, whereas the right hand-side to the one-hot representation of the thematic role played by the filler.

Figure 1: The architecture of the SG model, with the **update network** on the left hand-side and the **query network** on the right hand-side.

that the SG model maps from linguistic input to event meaning without any inbuilt knowledge of syntactic rules.

## 2 The Sentence Gestalt model

The SG model consists of an update and a query network (Fig. 1). The **update network** sequentially processes each incoming word to update activation of the SG layer, which represents the meaning of the sentence after the presentation of each word as a function of its previous activation and the activation induced by the new incoming word. It is composed of an input layer, which generates a vectorial representation $\vec{w}_t$ for each input word $i_t$ of the incoming sentence, and a LSTM recurrent layer generating a SG representation $\vec{sg}_t$ as a function of $\vec{w}_t$ and previous gestalt $\vec{sg}_{t-1}$ (Hochreiter and Schmidhuber, 1997). The **query network**, instead, extracts information concerning the event described by the sentence from the activation of the SG layer. It is composed by an hidden layer $\vec{h}_t$ combining the SG vector $\vec{sg}_t$ and a probe vectors $\vec{p}_i$, and an output layer generating a role-filler vector $\vec{o}_i$ from the hidden state $\vec{h}_t$.

The representation of the event described by a sentence consists of a set of role-filler vectors $\vec{o}_i$, each of which consists of the concatenation of the feature representation of a word and a one-hot vector of the role of that word in the context of the event described by the sentence (Fig. 2.a)).

During **training**, the model is presented with sentences, word by word and it is probed concerning the complete event, even if the relevant information has not yet been presented at the input layer. Crucially, no explicit information concerning the syntactic structure of the sentence is provided, nor any parsing process is explicitly implemented into the model. A probe consists of a vector $\vec{p}_i$ of the same size of a corresponding role-filler vector $\vec{o}_i$, but with either the thematic role identifier zeroed (Fig. 2.b) – if probing for roles –, or filler features zeroed (Fig. 2.c) – if instead probing for fillers. Responding to a probe consists therefore of completing the role-filler vector. Fillers are represented using word embeddings obtained by binarizing *Fasttext*. The discrepancies between the observed role-filler vector $\vec{o}_i$ and generated output $\vec{\hat{o}}_i$ is computed using cross-entropy and is back-propagated through the entire network to adjust its parameters in order to minimize the difference between model-generated and correct output.

## 3 Materials and methods

### 3.1 Training corpus and hyper-parameters

The SG model was trained on the British National Corpus section of the Rollenwechsel-English (RW-eng) corpus (Sayeed et al., 2018). The RW-eng corpus is annotated with semantic role information based on PropBank roles (Palmer et al., 2005) representing the event described by the sentence as a predicate and its arguments and modifiers. The SG model is trained on mapping each RW-eng sentence to its PropBank-style event representation.

The parameters of the SG model were optimized using Adamax (learning rate 0.0005) and mini-batches of size 32. Training was conducted for a maximum of 100 epochs on 90% of the batches, the remaining 10% was kept for validation (10 randomly initialized SG models were trained for the present simulations).

The size of the hidden layers (including the SG layer) was 600, whereas the input layer generates per-word embeddings of size 300 for the 10000 word forms accepted. The probe and output layers

had size 337 due to the concatenation of the 300-size binarized embedding vector, the frame number and the argument type.

### 3.2 Stimuli

Stimuli consisted of 360 sentences split in 4 matched conditions (2 active and 2 passive, with 90 sentence per condition). Conditions consist of control (C) and reversal anomaly (RA), both active and passive. RA sentences were generated starting from each C sentence. A RA sentence is obtained by reversing agent and patient fillers of a C sentence. So, for instance, C sentence "After decades in the jungle the research identified the species" is matched by RA "After decades in the jungle the species identified the research".

## 4 Role accuracy

After feeding the SG model with a whole sentence, the model is tested whether it correctly recognises the semantic role of the sentence's arguments by providing probes containing only the embeddings representing the agent or patient filler. No role information is provided by such probes. Role predictions are considered correct if the output role-filler vector contains a representation of agent role for agent fillers, or patient role for patient fillers. For instance, given the sentence "After decades in the jungle the research identified the species", the model estimate is correct if after being probed with filler "research" the output indicates *agent*, and when after being probed with filler "species" the role-filler output indicates *patient*.

Table 1 contains the role accuracy confusion matrices split in the four tested conditions averaged across 10 models. There was a significant main effect of condition, with significantly higher accuracies for C as compared to RA sentences ($F(1, 32) = 3212.0$, $p < 0.001$) and a main effect of voice, with significant higher accuracies for active as compared to passive sentences ($F(1, 32) = 113.5$, $p < 0.001$). There also was a statistically significant interaction between condition and voice in the average role accuracies of the SG models ($F(1, 32) = 299.7$, $p < 0.001$). In the RA condition, the SG models shows strong tendency to misinterpret *agents* as *patients* 88.27% of times for active and 81.23% of the times for passives. The rate of misinterpretation of *patients* as *agents* is lower, yet still significantly higher than in C sentences.

| | Ag | Pat | Prd | M* |
|---|---|---|---|---|
| **C active** | | | | |
| | Ag | Pat | Prd | M* |
| Ag | **91.98** | 6.91 | 0.00 | 1.11 |
| Pat | 1.60 | **91.98** | 2.59 | 3.83 |
| **RA active** | | | | |
| | Ag | Pat | Prd | M* |
| Ag | **4.81** | 88.27 | 2.59 | 4.32 |
| Pat | 48.27 | **50.12** | 0.00 | 1.60 |
| **C passive** | | | | |
| | Ag | Pat | Prd | M* |
| Ag | **44.57** | 51.36 | 0.00 | 4.07 |
| Pat | 1.60 | **90.62** | 2.84 | 4.94 |
| **RA passive** | | | | |
| | Ag | Pat | Prd | M* |
| Ag | **5.93** | 81.23 | 2.84 | 10.00 |
| Pat | 37.41 | **60.62** | 0.00 | 1.98 |

Table 1: Role probing confusion matrix for our four conditions. Rows indicate correct (target) roles, columns the percentage of correct (in bold) and misclassified fillers. **Ag** stands for *agent*, **Pat** for *patient*, **Prd** for *predicate*, and **M\*** for any other PropBank role. We included *patient*, *predicate*, and other roles because the SG model is free to assign any of the 27 different PropBank roles to a probed filler.

## 5 Filler accuracy

Fillers are predicted by feeding a whole sentence to the SG model together with the probe containing only the agent or patient role. No filler representation is provided by the probe. The model is expected to produce a role-filler vector containing the embedding representation of the correct filler for the probed role. Accuracy is computed by comparing the predicted filler embedding for a role to the correct embeddings of the sentence *agent* and *patient* fillers. If the predicted filler for the *agent* role is more similar – as cosine similarity – to the target embedding of the actual sentence *agent* filler as compared to the sentence *patient* filler, or vice versa, the prediction is considered correct. For instance, given the sentence "After decades in the jungle the research identified the species", the model prediction is correct if after being probed with role *agent* the output role-filler vectors is more similar to the embedding of "research" than to the embedding of "species"; and, conversely, when after being probed with role *patient* the role-filler output vector is more similar to the embedding of "species" than to the embedding of "research"

|     | C | | RA | |
| --- | --- | --- | --- | --- |
|     | active | passive | active | passive |
| Ag | 96.05 | 75.93 | 30.25 | 50.00 |
| Pat | 95.80 | 91.85 | 45.19 | 73.83 |
| avg. | 95.93 | 83.89 | 37.72 | 61.91 |

Table 2: Filler probing accuracy scores. Values are percentages. As in Table 1, **Ag** stands for *agent* and **Pat** for *patient*. We only included these two because the pairwise-similarity metric used to asses filler accuracy only consider *agent* and *patient* potential fillers.

Table 2 shows the filler accuracies across condition and voices averaged across 10 models. There was a significant main effect of condition, with significantly higher accuracies for C as compared to RA sentences ($F(1, 32) = 815.60$, $p < 0.001$) and a main effect of voice, with significantly higher accuracies for active as compared to passive sentences ($F(1, 32) = 18.75$, $p < 0.001$)). There also was a statistically significant interaction between condition and voice in the average filler accuracies of the SG models ($F(1, 32) = 166.54$, $p < 0.001$).

# 6 Conclusions

It has been reported that humans often misinterpret the agent and patient of reversal anomaly sentences such as e.g., "The dog was bitten by the man". This observation has offered the ground to the "good enough" theory of language comprehension, which assumes that role-filler assignment might sometimes rely on heuristics based on expectations about events, and is not always in line with the syntactic structure of the sentence (Ferreira et al., 2002; Ferreira, 2003).

In this paper, we show that the SG model, a simple connectionist model of sentence comprehension that is trained on mapping sequences of words to event representations based on a large scale corpus, displays similar biases as humans when it comes to comprehend control and reversal anomaly sentences. Despite the simple architecture and the lack of explicit syntactic training, it performs well in identifying roles and fillers for canonical active sentences. Its performance degrades somewhat for sentences in the passive voice and strongly degrades for RA sentences, syntactically correct but semantically anomalous sentences whose agent and patient fillers are swapped.

The model thus provides a computationally explicit account of plausibility based comprehension, which has posed a challenge to classic linguistic theories and models.

# Acknowledgements

The research presented in this paper was supported by the German collaborative research centre SFB1294 "Data Assimilation" and the Emmy Noether grant RA 2715/2-1. We thank ***removed for anonymity*** who helped with the creation of the stimuli and with the analyses.

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
