# OpenReview forum: "Plausibility based comprehension in a neural network model of sentence processing"
_aclweb.org/ACL/2022/Workshop/CMCL — Submitted to CMCL 2022_

### Official Review · Reviewer_Noc4 · 2022-03-16
**the perennial appeal of syntax-free comprehension**

**Rating:** 4
**Confidence:** 5

**Review:**

This paper represents a revival of the view originally championed by Tom Bever in 1970 that some route of the human sentence processing mechanism is free from the influence of grammar.  While Bever's own proposal includes a 2nd route that is influenced by grammar, this submission only focuses on the first route. The paper exhibits a model that faithfully misperceives Patients as Agents when they are more plausible in that semantic role.

In the writing, I sense a suggestion that somehow modeling this phenomenon bolsters the view that syntax is unnecessary (e.g. "pose a challenge to classic linguistic theories and models" page 4). This is not true. This view, while perennially-appealing to syntax-haters around the world, is simply not supported by Ferriera's results, Kuperberg's or by this paper's modeling. Rather what I think this paper shows is that recurrent neural networks can faithfully implement what David Caplan (MIT Press 1992) has called the "heuristic route" -- I take this to be roughly the same as Bever's syntax-free Perceptual Strategies from the 1970s.

If this submission's rhetoric were recalibrated to say for example "this is a model of what people with Broca's Aphasia can do" it could make a valuable contribution by deriving predictions about new cases where plausibility might override structure. Another improvement would be to characterize the contribution of the gated units (i.e LSTM) as compared to Doug Rohde's thesis which used ungated, Elman style Simple Recurrent Nets.

I am recommending reject because publication of this version would reflect badly on CMCL. As inclusive as we are at this workshop, scholarship is a non-negotiable requirement. However, if the author were to reframe the contribution i.e. to remove the spurious argument against classic linguistic theories and models, I think it would be appropriate to accept.

---

### Official Review · Reviewer_iUsJ · 2022-03-26
**The paper could be a bit more carefully worded**

**Rating:** 6
**Confidence:** 4

**Review:**

This paper describes a neural role-filler model that correctly mixes up arguments in sentences manipulated to have inverted canonical subjects and objects.

I think there's a contribution here, but the motivation should be worded a bit more carefully.  Evidence of human errors on these kinds of manipulated stimuli may suggest an auxiliary 'good enough' comprehension model which runs in parallel with a more precise parser that can reliably comprehend complex sentences with nested conditions, conjunctions, negations, etc.  This would not challenge the view that parsers are necessary, as claimed in the first paragraph.  Statements like 'Most theories ...' in the introduction need some citations, as does the sentence after that (presumably that refers to Ferreira and Kutas, et al).

Also, a clarity issue: on line 125 there a reference to 'Fasttext' which isn't explained.

A minor point: swapped arguments is also a kind of speech error; I wonder if comprehension errors are simply assuming a speech error has happened?

Finally, the acknowledgements section is usually omitted in anonymous submissions, as it has the potential to de-anonymize the submission.

---

### Official Review · Reviewer_GMha · 2022-03-27
**Misses the point I'm afraid**

**Rating:** 3
**Confidence:** 4

**Review:**

This paper addresses the phenomenon of human comprehenders frequently misassign thematic roles to arguments in ways that seem to be driven by "processing heuristics based on their expectations about events ... rather than relying on syntactic rules". For example, participants frequently misinterpret 'The dog was bitten by the man' as having the man as the patient and the dog as the agent. The authors train a neural network model to construct a representation of a sentence which captures information about the assignment of thematic roles to arguments. They find that, in line with what has been observed in humans, the model's accuracy in assigning roles is lowest for passive, semantically-anomalous sentences like 'The dog was bitten by the man', and highest for active, non-anomalous sentences like 'The dog bit the man'.

This would be somewhat interesting if the only empirical facts at hand were the facts about misinterpretations that the authors review in the introduction. (There would be the usual questions about the interpretability of the neural model, but there would at least be something to talk about.) But the whole interest of the Ferreira-style findings is the fact that those interpretations are *mis*interpretations -- when we say that they are *mis*interpretations, we don't mean that they diverge from what the stodgy old syntax textbooks say, we mean that they diverge from what those same human comprehenders will tell you if you give them a couple of extra seconds to think about it. That's the other fact at hand which the authors seem to completely disregard (and that's why the stodgy old syntax textbooks say what they say). The whole interest, in other words, is what sort of mind would exhibit this divergence between interpretations at two different timescales. The model described in the paper does not exhibit any such divergence, it only captures one half of the picture.

---

### Decision · Program_Chairs · 2022-03-29

Reject